# Experimental Investigation of the Different Polyacrylamide Dosages on Soil Water Movement under Brackish Water Infiltration

**DOI:** 10.3390/polym14122495

**Published:** 2022-06-19

**Authors:** Jihong Zhang, Quanjiu Wang, Weiyi Mu, Kai Wei, Yi Guo, Yan Sun

**Affiliations:** 1State Key Laboratory of Eco-Hydraulics in Northwest Arid Region of China, Xi’an University of Technology, Xi’an 710048, China; zhangjihong_eric@163.com (J.Z.); 18291869766@163.com (K.W.); 15503635823@163.com (Y.G.); 11414014@zju.edu.cn (Y.S.); 2College of Water Resources and Architectural Engineering, Shihezi University, Shihezi 832000, China; 3Key Laboratory of Modern Water-Saving Irrigation of Xinjiang Production and Construction Corps, Shihezi University, Shihezi 832000, China

**Keywords:** polyacrylamide, brackish water, sandy loam soil, soil water infiltration, soil hydraulic parameters

## Abstract

The use of soil conditioners in conjunction with brackish water irrigation is critical for the efficient development and use of brackish water as well as the enhancement of the structure of saline soil and stimulating crop growth. This study investigated the effects of different polyacrylamide (PAM) dosages (0, 0.02%, 0.04%, and 0.06%) on the water flow properties of sandy loam during brackish water infiltration using one-dimensional vertical and horizontal soil column infiltration experiments. The results showed that: (1) PAM could lower the soil infiltration rate and increase soil water retention performance under brackish water infiltration conditions. (2) PAM had a significant effect on the parameters of the Philip and Kostiakov infiltration models. The soil sorption rate *S* and the empirical coefficient *λ* were the smallest, and the empirical index *β* was the largest when the PAM dosage was 0.04%. (3) PAM dosage displayed a quadratic polynomial connection with the soil saturated water content and the saturated hydraulic conductivity. The soil saturated water content was highest when the PAM dosage was 0.04%, the intake suction *h_d_* of the Brooks-Corey model increased by 15.30%, and the soil water holding capacity was greatly improved. (4) Soil treated with PAM could absorb more water under the same soil water suction, whereas the soil unsaturated hydraulic conductivity and its growth rate decreased. The soil saturated diffusion rate *D_s_*, as well as the soil water diffusion threshold, rose. Finally, the 0.04% PAM dosage could improve soil hydrodynamic characteristics under brackish water infiltration, which is beneficial for the efficient utilization of brackish water.

## 1. Introduction

The supply of fresh water resources has become severely insufficient as the social economy has developed, and the gap between the supply of and demand for fresh water resources is becoming increasingly apparent [1]. Irrigation with brackish water is a significant measure to address the scarcity of irrigation water resources [2,3]. While brackish water irrigation can provide the necessary water for agricultural growth, it also introduces salt into the soil, causing salt to build to various degrees and impairing crop growth [4,5,6]. As a result, maximizing the use of brackish water resources while maintaining soil quality, avoiding soil degradation, and preventing declining land productivity has become a bottleneck challenge for agricultural sustainable development in northwest arid areas, such as Xinjiang, China [7,8].

Previous studies on the effects of brackish water irrigation mixed with soil conditioners on soil structure have offered theoretical directions for soil improvement [9,10]. Polyacrylamide (PAM, (C_3_H_5_NO)_n_) has good water solubility, flocculation, and chemical activity, and thus it has been explored and used as a soil structure modifier all over the world [11,12,13]. Kebede et al. (2022) indicated that PAM could improve soil structure stability and reduce runoff and soil erosion [14]. Wang et al. (2021) found that PAM could enhance soil wind erosion resistance [15]. Soltani et al. (2021) pointed out that PAM could be employed in expansive soils in South Australia [16]. Lentz and Sojka (1994) showed that PAM had an extremely high water absorption and retention capacity [17]. Cao et al. (2008) investigated the effect of PAM on water-stable aggregates of several soil types on the Loess Plateau and found that PAM improved the soil structure and increased the number of soil macroaggregates [18]. Feng et al. (2008) analyzed the effect of PAM on soil evaporation, indicating that PAM addition in the range of 0–2 g/m^2^ could reduce soil bulk mass, enhance soil water absorption and release capacity, and inhibit soil evaporation [19]. Han et al. (2010) studied the effects of PAM on soil physical properties and water distribution and found that PAM might improve soil water retention and water holding capacity [20].

A considerable number of investigations on soil water flow characteristics under the condition of brackish water infiltration have been performed [21,22,23]. Shi et al. (2007) compared and analyzed the parameters of the Philip model and Green-Ampt model under the conditions of brackish water infiltration through a vertical one-dimensional infiltration experiment [24]. Bi et al. (2010) conducted a comparative analysis and research on the infiltration characteristics of fresh water and brackish water [25]. Wang et al. (2014) studied the infiltration lows of sandy saline-alkali soil with chemical amendments under fresh water infiltration conditions to better understand the impact of PAM and other additives on soil infiltration characteristics [26].

However, there is still a scarcity of studies on the use of brackish water and PAM in combination. Under brackish water infiltration, the internal mechanism of the PAM dosage on soil water movement characteristics is still unknown. As a result, the effects of different PAM dosages on the water flow characteristics of sandy soil under brackish water infiltration conditions were discussed in this paper using one-dimensional vertical and horizontal soil column infiltration experiments, and the water distribution characteristics under different PAM dosages were analyzed so as to provide a theoretical basis for the rational use of brackish water in arid areas.

## 2. Materials and Methods

### 2.1. Tested Soil and Water Samples

The tested soil samples were collected from the 0–20 cm soil layer of the experimental field (86°10′ N, 41°35′ E) at the Bazhou Water Conservancy Administration Experimental Station in Xinjiang, China. The bulk density was determined using the ring knife method (1.63 g/cm^3^), and the recovered soil samples were air-dried before being utilized as a backup through a 2 mm sieve [27]. The mechanical composition was determined using a laser particle size analyzer (Mastersizer 2000, Marvin Instruments Co., Ltd., London, UK). The volume fractions of clay, silt, and sand in sandy loam were 2.94%, 32.54%, and 64.52%, respectively. The soil saturated water content and initial water content were 0.3879 and 0.0078 cm^3^/cm^3^, respectively, and the initial salt content of the soil was 3.35 g/kg. The brackish water for the experiment was drawn from the station’s subterranean well, which had a salinity of 2.01 g/L, and the contents of eight major ions, HCO_3_^−^, CO_3_^2−^, SO_4_^2−^, Cl^−^, Ca^2+^, Mg^2+^, Na^+^, and K^+^, were 15.02, 4.74, 2.41, 5.25, 0.94, 0.65, 11.07, and 2.01 mmol/L, respectively.

### 2.2. Experimental Methods

The experiments included the tone-dimensional vertical soil column ponding infiltration test and the one-dimensional horizontal soil column infiltration test. The vertical soil column was 8 cm in diameter and stood 60 cm tall, while the horizontal soil column was 8 cm in diameter and measured 60 cm in length. The organic glass floor at the bottom of the dirt column was 0.5 cm thick, with 0.2 cm pores for exhaust. On the side wall, there were soil holes every 2.5 cm with a diameter of 1.5 cm, which was convenient for soil water. A Mahalanobis bottle with a 50 cm^2^ cross-sectional area and a 60 cm height was used to provide a consistent water head in the water delivery system.

Each 5 cm layer was layered into the soil column according to the soil bulk density of 1.63 g/cm^3^. The application mode of PAM was a mixed application (i.e., mixed application of PAM and dry soil), and the application rates of PAM were 0, 0.02%, 0.04%, and 0.06% according to the dry soil mass ratio. PAM was initially mixed with the needed dry soil to ensure that it had the desired effect, and the PAM was sealed with plastic film and left in the room for 12 h after mixing with a certain amount of water in the spray kettle. Then, PAM was fitted at the necessary soil column position after drying.

The water depth in the vertical soil column test was kept at around 3 cm, and the water chamber length in the horizontal infiltration test was 10 cm. The water level in the Mahalanobis bottle and the distance of the wetting front from the surface of the soil column were measured using the dense first approach and sparse later approach during the test. When the wetting front reached the regulated depth (a vertical wetting front of 35 cm and a horizontal wetting front of 37 cm), the water supply was turned off, and the accumulated water was promptly evacuated from the water chamber. Rapid discharge of accumulated water with filter paper to dry the surface water, and from the side wall of the soil column hole with small soil drill extraction samples. The soil water content was determined using the drying method (105 ± 2 °C) [28].

The soil saturated water content was determined using the ring knife method under various PAM dosages, and each experiment was repeated three times [22]. The soil saturated hydraulic conductivity under different PAM dosages was measured by means of the constant head method [23]. A short organic glass soil column with an 8 cm diameter and a 20 cm height was chosen. To prevent soil particles from obstructing the outflow, gauze and filter paper were loaded at the bottom of the dirt column. The soil column was also placed into the soil column using a 1.63 g/cm^3^ soil bulk density and a soil height of 10 cm. The soil column was first soaked with brackish water and then the water head was set to around 3 cm. The outflow valve was opened, and the amount of seepage water over a certain time period was measured. Each test was carried out three times in total.

### 2.3. Basic Theory

On the basis of the accumulated water infiltration test, Philip (1969) solved the basic equation of soil moisture movement by power series and obtained the Philip infiltration model [29]:(1)$$I=St^{0.5}$$
where *I* is the cumulative infiltration (cm), *S* is the soil permeability (cm/min^0.5^), and *t* is the infiltration time (min).

The specific expression of Kostiakov infiltration model is as follows [30]:(2)$$I=\lambda t^{1-\beta }$$
where *β* is the empirical infiltration index, which reflects the attenuation rate of soil infiltration capacity. *λ* is the empirical infiltration coefficient, which represents the cumulative infiltration volume at the end of the first unit period after the beginning of infiltration and is numerically equal to the average infiltration rate of the first unit period (cm/min).

Soil water dynamic parameters are the basis for simulating and predicting soil water movement. The expression of soil water characteristic curve and soil unsaturated hydraulic conductivity proposed by Brooks-Corey (1964) is as follows [31]:(3)$$U=\frac{\theta -\theta_{r}}{\theta_{s}-\theta_{r}}={\left(\frac{h_{d}}{h}\right)}^{n}$$
(4)$$K(U)=K_{s}{\left(\frac{h_{d}}{h}\right)}^{m}=K_{s}U^{\frac{m}{n}}$$
where *U* is the soil effective saturation, while *θ*, *θ_s_*, and *θ_r_* are the soil water content, saturated water content, and residual water content (cm^3^/cm^3^), respectively. The residual water content is equal to the initial water content when the initial soil moisture content is low. *K*(*U*) is the soil unsaturated hydraulic conductivity (cm/min), and *K_s_* is the soil saturated hydraulic conductivity (cm/min). *h* is the soil suction (cm), and *h_d_* is the intake suction (cm). *n* is the shape coefficient, and *m* is the empirical coefficient, m = 3*n* + 2.

Wang et al. (2002) proposed a method to calculate the parameters of the Brooks-Corey model based on the horizontal permeability test data [32]:(5)$$n=\sqrt{\frac{\theta_{s}-\theta_{r}}{A_{1}+\theta_{i}-\theta_{r}}-1}$$
(6)$$h_{d}=\frac{A_{2}}{anK_{s}}$$
where *a* is the parameter, which is approximately 1 when the initial soil content is very small.

*A*_1_ and *A*_2_ can be obtained from the cumulative infiltration, wetting front, and time data in the horizontal infiltration process.
(7)$$I=A_{1}x_{f}$$
(8)$$i=A_{2}/x_{f}$$

Wang et al. (2004) suggested a simple method to calculate the soil unsaturated water diffusivity according to the horizontal infiltration test data [33]:(9)$$D(S)=D_{s}S^{L}$$
(10)$$D_{s}=\frac{A_{1}A_{2}}{\left(\theta_{s}-\theta_{i})(\theta_{s}-\theta_{i}-A_{1}\right)}$$
(11)$$L=\frac{A_{1}}{\theta_{s}-\theta_{i}-A_{1}}-1$$
where *D_s_* is the soil saturated water diffusivity (cm^2^/min), and *L* is the parameter.

### 2.4. Statistical Analysis

All measured data were recorded in Excel 2019 and assessed by means of analysis of variance (ANOVA) using SPSS 22.0 software (IBM Corp., Armonk, NY, USA). Significant differences (*p* < 0.05) between means were identified using the least significant difference (LSD) test. Figures were drawn using Origin 2021 software (OriginLab Corporation, Northampton, MA, USA).

## 3. Results and Discussion

### 3.1. Effect of PAM Dosages on Infiltration Characteristics of Brackish Water

The variation process of the cumulative infiltration and wetting front of soil treated with PAM over time under the condition of brackish infiltration is shown in Figure 1. The cumulative infiltration of brackish water and the rising depth of the wetting front under each PAM application rate exhibited a high degree of synchrony in the first 100 min of infiltration. Because PAM had a limited effect early in the infiltration process, the cumulative infiltration and wetting front had minimal difference [22]. The infiltration depth of soil grew as the time of infiltration increased, and PAM and soil interacted and played a full part, resulting in variances in the cumulative infiltration and increasing wetting front depth. After 100 min of infiltration, the cumulative infiltration and wetting front depth increase showed a tendency for reducing and then increasing, with an increase in PAM dosages and the same wetting front. Under a 0.04% PAM dosage, the time required to achieve the same infiltration depth was the longest. This is because PAM is a long-chain polymer compound that primarily impacts the viscosity of soil water [34,35]. With the increase in PAM dosage, the ability of PAM to bond to soil water improved, and the viscosity of soil water increased, resulting in a decrease in the soil water infiltration rate [11]. When the level of PAM application rate reached 0.06%, the soil cumulative infiltration increased again. According to Gungor and Karaoglan (2001), when the PAM dosage was too high, the existence of exchangeable Na^+^ reduced the viscosity of PAM aqueous solution, increasing the infiltration rate of soil water [36]. After infiltration, the average volumetric water contents of the wetting body were 0.2787, 0.3216, 0.3320 and 0.3135 cm^3^/cm^3^, respectively, for 0%, 0.02%, 0.04%, and 0.06% PAM dosages. Compared with the wetting body without PAM application, the volumetric water contents of the wetting body with 0.02%, 0.04%, and 0.06% PAM application increased by 15.40%, 19.13%, and 12.52%, respectively. When the PAM dosage was 0.04%, the water retention effect was the best. This is because PAM improves soil structure [12], causes dispersed large particles in the soil to bond [37], promotes the formation of soil aggregates [13], increases soil porosity [35], lowers the soil infiltration rate [23], makes water infiltration more uniform, and ensures that more water is retained in the soil layer, all of which are important for root water absorption and sandy loam soil water retention.

### 3.2. Effect of PAM Dosages on Soil Water Distribution

Figure 2 depicts the fluctuation in soil water content with depth under various PAM dosages. Overall, the water content of the surface soil was the highest, approaching saturation, the water content of the wetting front was the lowest, and the water content below the wetting front was close to the initial water content. The soil water content increased initially and then dropped as the PAM dosage was raised. The soil water content was highest when the PAM dosage was 0.04%.

The soil water holding efficiency in the process of infiltration was defined as the ratio of the difference between the water content and the control water content in a certain soil depth under PAM treatment. Table 1 shows the water holding capacity of each soil depth. The soil water holding efficiency rose with increasing soil depth, peaking at 20–30 cm. With varying PAM dosages, the soil water holding efficiency varied, and the maximum soil water holding efficiency was found at the 0.04% PAM dosage, with a depth of 20–30 cm yielding a water holding efficiency of 28.36%.

### 3.3. Effect of PAM Dosages on Infiltration Model Parameters

The Philip and Kostiakov infiltration models were used to fit the infiltration data on the basis of the measured data (Table 2). These two infiltration models had a good fitting effect, and the determination coefficient *R*^2^ reached more than 0.98. The association between the PAM dosages and infiltration parameters was investigated further. In the Philip model, as the PAM dosage increased, the sorption rate *S* first declined and subsequently climbed. When the PAM dosage was 0.04%, the sorption rate *S* dropped to 0.547 cm/min^0.5^, showing that the capillary force’s ability to absorb water in soil was decreased. The reason for this could be that the hydrogel generated when PAM was introduced to the soil increased the viscosity of water, decreasing the capillary force’s water absorption capacity [11,34]. The capillary force’s water absorption capacity in soil was lowest when the PAM application rate was 0.04%. The empirical coefficient *λ* reduced first and then grew in the Kostiakov model when the PAM dosage increased, while the empirical index *β* climbed first and then decreased. The empirical coefficient *λ* was the smallest, and the empirical index *β* reached the maximum when the PAM dosage was 0.04%, indicating that the initial infiltration rate of soil was the smallest and the soil infiltration capacity was the smallest when the PAM dosage was 0.04 percent. According to a study by López-Maldonado et al., PAM or other polymers could effectively promote the coagulation of soil colloid, increase the number of soil aggregates, and improve soil structure [38].

### 3.4. Effects of PAM Dosages on Soil Saturated Water Content and Saturated Hydraulic Conductivity

The soil saturated water content *θ_s_* is a significant soil water constant that might reflect the water retention ability of the soil, and the soil saturated hydraulic conductivity *K_s_* is a crucial indicator for simulating soil water movement because it reflects soil hydraulic conductivity. The relationship between soil saturated water content, saturated hydraulic conductivity, and PAM dosage under the condition of brackish water infiltration is shown in Figure 3. The soil saturated water content increased first and then decreased as the PAM dosage increased, which was primarily due to the use of PAM to change the soil structure by increasing the number of small pores, total soil pores, and soil water absorption capacity, which resulted in an increase in soil saturated water content [11,37]. As a result, using PAM in the soil could help to improve the ability of soil water and fertilizer. The binomial equation was used to fit the soil saturated water content (*θ_s_*, cm^3^/cm^3^) to the PAM dosages (*P*, %). The fitting equation was *θ_s_* = −19.688 *P*^2^ + 1.4257 *P* + 0.3879, and the determination coefficient *R*^2^ was 0.997, which was statistically significant (*p* < 0.01). Furthermore, as the PAM dosage was raised, the soil saturated hydraulic conductivity first declined and subsequently increased. The main reason for this is that when PAM is applied to the soil, the number of small pores grows, the number of large pores drops, and the soil infiltration capacity reduces, lowering the saturated hydraulic conductivity of the soil [9,21]. The binomial equation was used to fit the association between soil saturated hydraulic conductivity (*K_s_*, cm/min) and the PAM dosages (*P*, %). The fitting equation was *K_s_* = 4.149 *P*^2^ − 4.394 *P* + 2.233, and the determination coefficient *R*^2^ was 0.885, which reached a significant level (*p* < 0.05).

### 3.5. Effect of PAM Application Rate on Parameters of Brooks-Corey Model

The coefficients *A*_1_ and *A*_2_ were fitted by Equations (7) and (8) using the data from the horizontal infiltration test (Table 3). The fitting result was satisfactory, with a determination coefficient greater than 0.95. The coefficient *A*_1_ of soil treated with PAM was higher than that of the control treatment; however, the coefficient *A*_2_ was lower. At the same infiltration distance, the cumulative infiltration of soil treated with PAM was more than that of the control, while the infiltration rate was lower, which was consistent with the variance of cumulative infiltration and wetting front.

The shape coefficient *n*, the intake suction *h_d_*, and the empirical coefficient *m* were calculated by substituting the above fitting *A*_1_, *A*_2_ and the measured saturated hydraulic conductivity *K_s_* into Equations (5) and (6) (Table 4). As the PAM dosage was raised, the soil saturated water content increased at first and then declined, and it was highest when the PAM dosage was 0.04%, increasing by 6.47% over the control. This occurred because once PAM was added to the soil, the tiny particles were aggregated into larger aggregates, increasing the soil porosity and thus the saturated water content [14,23]. *h_d_* is the intake suction in the Brooks-Corey model, which is the crucial suction value of soil drainage. The greater the soil water holding capacity, the stronger the intake suction [39,40]. The intake suction *h_d_* rose first and then decreased when the PAM dosage was raised. The intake suction increased by 25.97% when the PAM dosage was 0.04% compared with the control, and the soil water holding capacity was greatly improved. The shape coefficient *n* was larger than that of the control treatment, except at the 0.04% PAM dosage, when it was lower than the control, although the association between the shape coefficient *n* and PAM dosage was not significant. PAM had little effect on the coefficient *m*, which had an average value of roughly 3.5.

The shape coefficient *n*, the intake suction *h_d_*, and the empirical coefficient *m* were substituted into Equations (2) and (3) to obtain the soil water characteristic curve and unsaturated hydraulic conductivity curve with different PAM dosages in order to clearly show the hydrodynamic characteristics of soil with PAM applied (Figure 4). With increasing soil water content, soil water absorption reduced significantly. The PAM-treated soil water characteristic curve was steeper than that of the control, indicating that the same soil water absorption could absorb more water. The soil water content of 0.04% PAM dosage increased by 27.52% when the soil water suction was 800 cm H_2_O, which was consistent with the distribution of soil water content after infiltration, indicating that PAM can also increase the soil water suction to a degree when the soil texture is the same. With the rise in soil water content, the unsaturated hydraulic conductivity of the soil increased rapidly. The unsaturated hydraulic conductivity of soil treated with PAM was lower than that of the control before saturation, and its growth rate was likewise lower than that of the control.

### 3.6. Effect of PAM Dosages on Soil Water Diffusivity

The soil saturated water diffusivity *D_s_* and parameter *L* were calculated by substituting *A*_1_ and *A*_2_ into Equations (10) and (11). With an increasing PAM application rate, the soil saturated water diffusivity *D_s_* fell at first, then increased (Table 5). The soil saturated water diffusivity *D_s_* reduced by 39.4% when PAM was applied at a rate of 0.04% compared with the control.

The saturated diffusion rate *D_s_* and parameter *L* were inversely obtained and substituted into Equation (9) to obtain the unsaturated diffusion rate of soil with varied PAM dosages (Figure 5). Soil water began to spread only after a particular threshold of moisture content was met, and the soil diffusion rate increased gradually as the water content rose. The unsaturated diffusion rate was much larger at a high water content than at a low water content. The water diffusion threshold of PAM-treated soil was higher than that of the control, and the water diffusion threshold of 0.04% PAM-treated soil increased by 10.82%. This is due to the fact that water infiltrating into the soil must first meet the membrane water absorbed on the surface of soil particles, then penetrate the fine pores of the soil, and finally become free water to spread forward in the process of soil water absorption and infiltration [41,42]. PAM has the ability to boost the number of tiny pores in soil [9,11], and the soil water can only diffuse forward when the small pores are fully filled and the soil water content is high, which explains why the wetting front depth of the soil treated with PAM was less than that of the soil without treated PAM.

## 4. Conclusions

The soil sorption rate *S* and the empirical coefficient *λ* were the smallest and the empirical index *β* was the largest when the PAM dosage was 0.04%. PAM dosage displayed a quadratic polynomial connection with soil saturated water content and saturated hydraulic conductivity. The soil saturated water content was highest when the PAM dosage was 0.04%, the intake suction *h_d_* of the Brooks-Corey model increased by 15.30%, and the soil water holding capacity was greatly improved. Soil treated with PAM could absorb more water under the same soil water suction, whereas the soil unsaturated hydraulic conductivity and its growth rate decreased. The soil saturated diffusion rate *D_s_*, as well as the soil water diffusion threshold, rose. Finally, a 0.04% PAM dosage could improve soil hydrodynamic characteristics under brackish water infiltration, which is beneficial for the efficient utilization of brackish water.

## Figures and Tables

**Figure 1 polymers-14-02495-f001:**
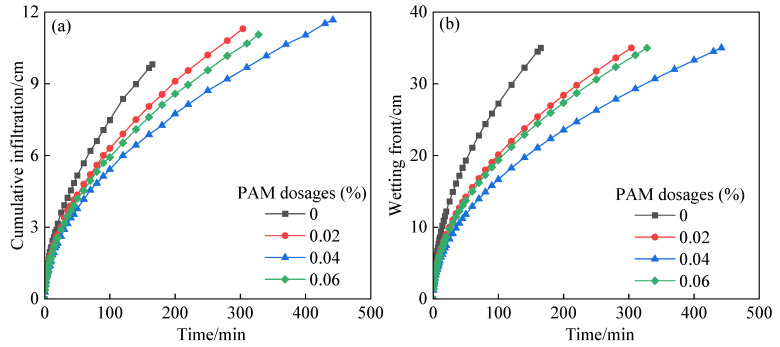
Effect of PAM dosages on soil infiltration characteristics of brackish water. (**a**) cumulative infiltration, (**b**)wetting front.

**Figure 2 polymers-14-02495-f002:**
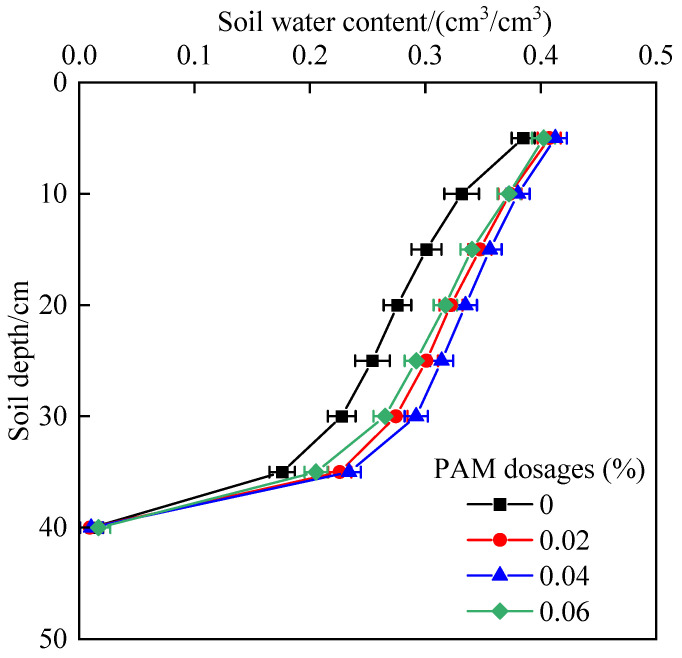
Effect of PAM dosages on soil water content distribution.

**Figure 3 polymers-14-02495-f003:**
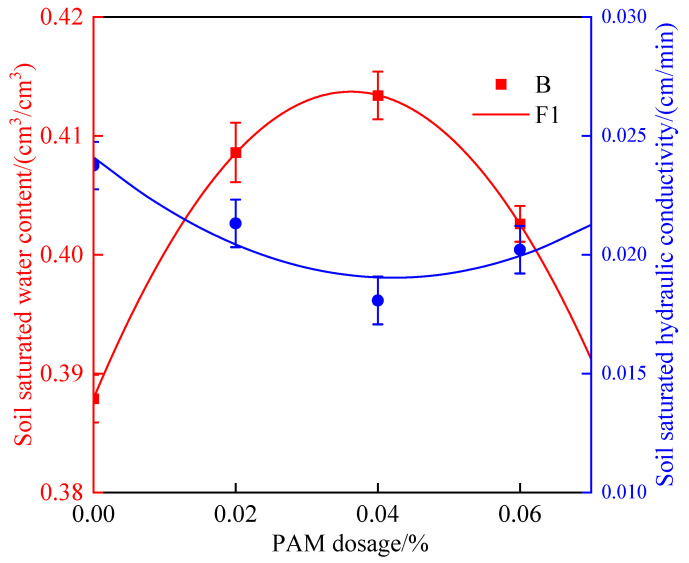
Effects of PAM dosages on soil saturated water content and saturated hydraulic conductivity.

**Figure 4 polymers-14-02495-f004:**
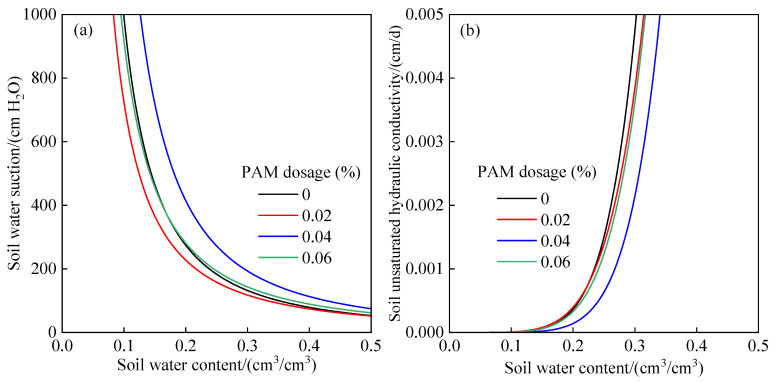
Soil water characteristic curve and unsaturated hydraulic conductivity curve under different PAM dosages. (**a**)soil water characteristic curve, (**b**) unsaturated hydraulic conductivity curve.

**Figure 5 polymers-14-02495-f005:**
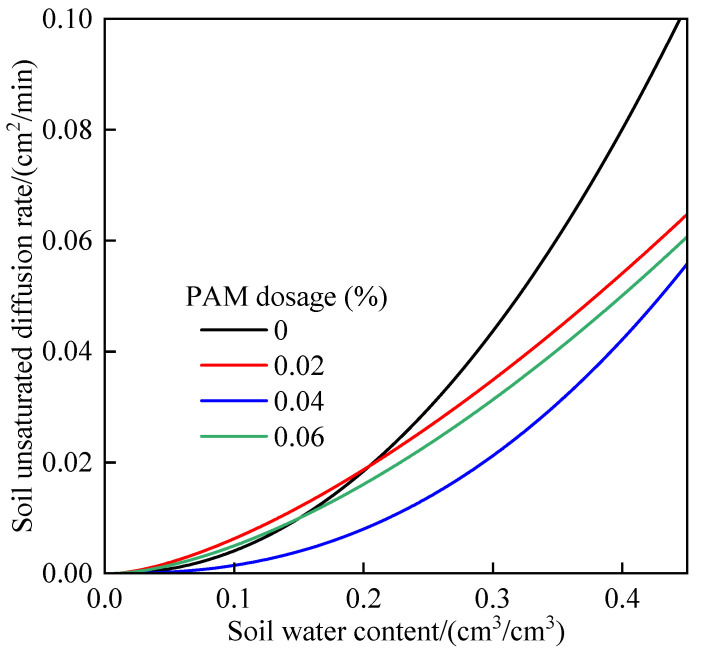
Soil unsaturated water diffusion rate under different PAM dosages.

**Table 1 polymers-14-02495-t001:** Soil water holding efficiency (%) under different PAM dosages.

Soil Depth (cm)	PAM Dosages (%)		
	0.02	0.04	0.06
0–10	12.59	14.72	12.36
10–20	16.82	21.37	15.04
20–30	20.66	28.36	16.50

**Table 2 polymers-14-02495-t002:** Infiltration model parameters under different PAM application dosages.

PAM Dosages	Philip Model		Kostiakov Model		
	Soil Sorption Rate *S* (cm/min^0.5^)	Determination Coefficient *R*^2^	Empirical Coefficient *λ*	Empirical Index *β*	Determination Coefficient *R*^2^
0	0.742	0.986	0.625	0.460	0.985
0.02%	0.635	0.985	0.561	0.475	0.986
0.04%	0.547	0.988	0.502	0.484	0.983
0.06%	0.600	0.984	0.532	0.476	0.984

**Table 3 polymers-14-02495-t003:** Fitting results of the coefficients *A*_1_ and *A*_2_.

Formula	Parameter	PAM Dosages (%)			
		0	0.02	0.04	0.06
*I* = *A*_1_*x_f_*	*A* _1_	0.287	0.286	0.312	0.285
	*R* ^2^	0.994	0.996	0.997	0.996
*i* = *A*_2_/*x_f_*	*A* _2_	1.066	0.682	0.634	0.640
	*R* ^2^	0.996	0.995	0.995	0.994

**Table 4 polymers-14-02495-t004:** Parameters of the Brooks-Corey model relative to the PAM amendment rate.

**PAM Dosages (%)**	**Parameters**		
	**Intake Suction *h_d_***	**Shape Coefficient *n***	**Empirical Coefficient *m***
0	83.23	0.572	3.503
0.02	71.33	0.634	3.438
0.04	106.1	0.548	3.563
0.06	88.09	0.621	3.551

**Table 5 polymers-14-02495-t005:** Soil saturated water diffusivity *D_s_* and parameter *L* under different PAM dosages.

**Parameter**	**PAM Dosages (%)**			
	**0**	**0.02**	**0.04**	**0.06**
Soil saturated water diffusivity *D_s_*	0.075	0.056	0.046	0.051
Parameter *L*	2.061	1.491	2.333	1.596

## Data Availability

The data presented in this study are available on request from the corresponding author. The data are not publicly available due to the project not being complete.

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
