# Peer review of "Experimental Investigation of the Different Polyacrylamide Dosages on Soil Water Movement under Brackish Water Infiltration"

_polymers, 2022, doi:10.3390/polym14122495_

Round 1

Reviewer 1 Report

After reviewing the following manuscript "Experimental investigation of the different polyacrylamide dosages on soil water movement under brackish water infiltration", the authors must address the following comments:

1. The authors must justify the use of PAM for this study, currently the trend is to use biopolymers.

2. Why do the authors not consider the measurement of a parameter that relates the physicochemical properties of the soil and the PAM? Could it be zeta potential?

This reference could support in justifying the use of polymers for this study and how zeta potential measurements could complement their soil water infiltration performance.

10.1016/j.cej.2021.130210 3.

3.The authors must outline the mechanistic part of their study. How does PAM interact with water and soil?

4. What are the chemical characteristics of the PAM used? Molecular weight? 5. The authors must highlight the originality of their work, commenting in the introduction that PAM has already been addressed as a soil conditioner in previous studies. The conclusions should be improved to comment on this aspect.

Author Response

Response to Reviewer 1 Comments

Point 1: The authors must justify the use of PAM for this study, currently the trend is to use biopolymers.

Response 1: Previous studies have shown that PAM has great potential in improving soil structure. Lentz and Sojka (1994) showed that PAM had an extremely high water absorption and retention capacity [17]. Cao et al. (2008) investigated the effect of PAM on water-stable aggregates of several soil types on the Loess Plateau, and found that PAM improved soil structure, and increased the number of soil macroaggregates [18]…

Point 2: Why do the authors not consider the measurement of a parameter that relates the physicochemical properties of the soil and the PAM? Could it be zeta potential?

This reference could support in justifying the use of polymers for this study and how zeta potential measurements could complement their soil water infiltration performance.

10.1016/j.cej.2021.130210 3.

Response 2: According to the suggestions of the reviewer, relevant references and explanations have been supplemented.

Point 3: The authors must outline the mechanistic part of their study. How does PAM interact with water and soil?

Response 3: PAM is a long-chain polymer compound that primarily impacts the viscosity of soil water. PAM binds dispersed large particles in the soil, promotes the formation of soil aggregates, improves soil structure, and increases soil fine pores.

Point 4: What are the chemical characteristics of the PAM used? Molecular weight?

Response 4: Polyacrylamide (PAM, (C3H5NO)n) has good water solubility, flocculation, and chemical activity, which has been explored and used as a soil structure modifier all over the world [11–13].

Point 5: The authors must highlight the originality of their work, commenting in the introduction that PAM has already been addressed as a soil conditioner in previous studies. The conclusions should be improved to comment on this aspect.

Response 5: Previous studies have shown that PAM has great potential to improve soil structure. There is still a scarcity of studies on the use of brackish water and PAM in combination. Under brackish water infiltration, the internal mechanism of PAM dosage on soil water movement characteristics is still unknown. Our study main investigated the effects of different polyacrylamide (PAM) dosages on the water flow properties of sandy loam during brackish water infiltration. The conclusions have been revised.

Reviewer 2 Report

Dear Authors,

I think that the manuscript is basically publishable in its present form.

I only had minor issues to be targeted, see attached pdf with comments.

Regards, Reviewer X

Author Response

Response to Reviewer 2 Comments

Point 1: I think that the manuscript is basically publishable in its present form.

Response 1: Thanks for your affirmation. We have carefully checked and modified the manuscript, again.

Point 2: I only had minor issues to be targeted, see attached pdf with comments.

Response 2: We have revised the relevant issues.

Round 2

Reviewer 1 Report

Dear

The authors have addressed the comments point by point and their work has improved substantially.

Best regards